# Image as Set of Points

**Xu Ma**[1*]**, Yuqian Zhou**[2*]**,  Huan Wang**[1]**, Can Qin**[1]**, Bin Sun**[1]**, Chang Liu**[1]**, Yun Fu**[1]
[1]Northeastern University     [2]Adobe Inc.
{ma.xu1,wang.huan,qin.ca,sun.bi,liu.chang6}@northeastern.edu
yuqzhou@adobe.com    yunfu@ece.neu.edu

## Abstract

What is an image and how to extract latent features?

**Convolutional Networks** (ConvNets) consider an image as organized pixels in a rectangular shape and extract features via convolutional operation in local region; **Vision Transformers** (ViTs) treat an image as a sequence of patches and extract features via attention mechanism in a global range. In this work, we introduce a straightforward and promising paradigm for visual representation, which is called **Context Clusters**. Context clusters (CoCs) view an image as a set of unorganized points and extract features via simplified clustering algorithm. In detail, each point includes the raw feature (*e.g.*, color) and positional information (*e.g.*, coordinates), and a simplified clustering algorithm is employed to group and extract deep features hierarchically. Our CoCs are convolution- and attention-free, and only rely on clustering algorithm for spatial interaction. Owing to the simple design, we show CoCs endow gratifying interpretability via the visualization of clustering process. Our CoCs aim at providing a new perspective on image and visual representation, which may enjoy broad applications in different domains and exhibit profound insights. Even though we are not targeting SOTA performance, COCs still achieve comparable or even better results than ConvNets or ViTs on several benchmarks. Codes are available at: https://github.com/ma-xu/Context-Cluster.

## 1    Introduction

*The way we extract features depends a lot on how we interpret an image.* As a fundamental paradigm, Convolutional Neural Networks (ConvNets) have dominated the field of computer vision and considerably improved the performance of various vision tasks in recent years (He et al., 2016; Xie et al., 2021; Ge et al., 2021). Methodologically, ConvNets conceptualize a picture as a collection of arranged pixels in a rectangle form, and extract local features using convolution in a sliding window fashion. Benefiting from some important inductive biases like locality and translation equivariance, ConvNets are made to be efficient and effective. Recently, Vision Transformers (ViTs) have significantly challenged ConvNets' hegemony in the vision domain. Derived from language processing, Transformers (Vaswani et al., 2017) treat an image as a sequence of patches, and a global-range self-attention operation is employed to adaptively fuse information from patches. With the resulting models (*i.e.*, ViTs), the inherent inductive biases in ConvNets are abandoned, and gratifying results are obtained (Touvron et al., 2021).

Recent work has shown tremendous improvements in vision community, which are mainly built on top of convolution or attention (*e.g.*, ConvNeXt (Liu et al., 2022), MAE (He et al., 2022), and CLIP (Radford et al., 2021)). Meanwhile, some attempts combine convolution and attention together, like CMT (Guo et al., 2022a) and CoAtNet (Dai et al., 2021). These methods scan images in grid (convolution) yet explore mutual relationships of a sequence (attention), enjoying locality prior (convolution) without sacrificing global reception (attention). While they inherit the advantages from both and achieve better empirical performance, the insights and knowledge are still restricted to ConvNets and ViTs. Instead of being lured into the trap of chasing incremental improvements, we underline that some feature extractors are also worth investigating beyond convolution and attention. While convolution and attention are acknowledged to have significant benefits and an enormous influence on the field of vision, they are not the only choices. MLP-based architectures (Touvron

---

[*]Equal contribution

et al., 2022; Tolstikhin et al., 2021) have demonstrated that a pure MLP-based design can also achieve similar performance. Besides, considering graph network as the feature extractor is proven to be feasible (Han et al., 2022). Hence, we expect a new paradigm of feature extraction that can provide some novel insights instead of incremental performance improvements.

In this work, we look back into the classical algorithm for the fundamental visual representation, clustering method (Bishop & Nasrabadi, 2006). Holistically, we view an image as a set of data points and group all points into clusters. In each cluster, we aggregate the points into a center and then dispatch the center point to all the points adaptively. We call this design context cluster. Fig. 1 illustrates the process. Specifically, we consider each pixel as a 5-dimensional data point with the information of color and position. **In a sense, we convert an image as a set of point clouds and utilize methodologies from point cloud analysis (Qi et al., 2017b; Ma et al.,**

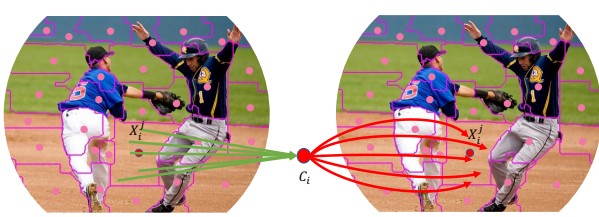

Figure 1: A context cluster in our network trained for image classification. We view *an image as a set of points* and sample $c$ centers for points clustering. Point features are aggregated and then dispatched within a cluster. For cluster center $C_i$, we first aggregated all points $\{x_i^0, x_i^1, \cdots, x_i^n\}$ in $i$th cluster, then the aggregated result is distributed to all points in the clusters dynamically. See § 3 for details.

**2022) for image visual representation learning.** This bridges the representations of image and point cloud, showing a strong generalization and opening the possibilities for an easy fusion of multi-modalities. With a set of points, we introduce a simplified clustering method to group the points into clusters. The clustering processing shares a similar idea as SuperPixel (Ren & Malik, 2003), where similar pixels are grouped, but they are fundamentally different. To our best knowledge, we are the first ones to introduce the clustering method for the general visual representation and make it work. On the contrary, SuperPixel and later versions are mainly for image pre-processing (Jampani et al., 2018) or particular tasks like semantic segmentation (Yang et al., 2020; Yu et al., 2022b).

We instantiate our deep network based on the context cluster and name the resulting models as Context Clusters (CoCs). Our new design is inherently different from ConvNets or ViTs, but we also inherit some positive philosophy from them, including the hierarchical representation (Liu et al., 2022) from ConvNets and Metaformer (Yu et al., 2022c) framework from ViTs. CoCs reveal distinct advantages. First, by considering image as a set of points, CoCs show great generalization ability to different data domains, like point clouds, RGBD images, *etc.* Second, the context clustering processing provides CoCs gratifying interpretability. By visualizing the clustering in each layer, we can explicitly understand the learning in each layer. Even though our method does not target SOTA performance, it still achieves on par or even better performance than ConvNets or ViTs on several benchmarks. We hope our context cluster will bring new breakthroughs to the vision community.

## 2 RELATED WORK

**Clustering in Image Processing** While clustering approaches in image processing (Castleman, 1996) have gone out of favor in the deep learning era, they never disappear from computer vision. A time-honored work is SuperPixel (Ren & Malik, 2003), which segments an image into regions by grouping a set of pixels that share common characteristics. Given the desired sparsity and simple representation, SuperPixel has become a common practice for image preprocessing. Naive application of SuperPixel exhaustively clusters (*e.g.*, via K-means algorithm) pixels over the entire image, making the computational cost heavy. To this end, SLIC (Achanta et al., 2012) limits the clustering operation in a local region and evenly initializes the K-means centers for better and faster convergence. In recent years, clustering methods have been experiencing a surge of interest and are closely bound with deep networks (Li & Chen, 2015; Jampani et al., 2018; Qin et al., 2018; Yang et al., 2020). To create the superpixels for deep networks, SSN (Jampani et al., 2018) proposes a differentiable SLIC method, which is end-to-end trainable and enjoys favorable runtime. Most recently, tentative efforts have been made towards applying clustering methods into networks for specific vision tasks, like segmentation (Yu et al., 2022b; Xu et al., 2022) and fine-grained recognition (Huang & Li, 2020). For example, CMT-DeepLab (Yu et al., 2022a) interprets the object queries in segmentation task as

cluster centers, and the grouped pixels are assigned to the segmentation for each cluster. Nevertheless, to our best knowledge, there is no work conducted for a general visual representation via clustering. We aim to make up for the vacancy, along with proving the feasibility numerically and visually.

**ConvNets & ViTs**   ConvNets have dominated the vision community since the deep learning era (Simonyan & Zisserman, 2015; He et al., 2016). Recently, ViTs (Dosovitskiy et al., 2020) introduce purely attention-based transformers (Vaswani et al., 2017) to the vision community and have set new SOTA performances on various vision tasks. A common and plausible conjecture is that these gratifying achievements are credited to the self-attention mechanism. However, this intuitive conjecture has soon been challenged. Extensive experiments also showcase that a ResNet (He et al., 2016) can achieve on par or even better performance than ViTs, with proper training recipe and minimal modifications (Wightman et al., 2021; Liu et al., 2022). We emphasize that while convolution and attention may have unique virtues (*i.e.*, ConvNets enjoy inductive biases (Liu et al., 2022) while ViTs excel at generalization (Yuan et al., 2021b)), they did not show significant performance gap. Different from convolution and attention, in this work, we radically present a new paradigm for visual representation using clustering algorithm. With both quantitative and qualitative analysis, we show that our method can serve as a new general backbone and enjoys gratifying interpretability.

**Recent Advances**   Extensive efforts have been made to push up the vision tasks' performances within the framework of ConvNets and ViTs (Liu et al., 2021b; Ding et al., 2022b; Wu et al., 2021). To take advantage of both convolution and attention, some work learns to mix the two designs in a hybrid mode, like CoAtNet (Dai et al., 2021) and Mobile-Former (Chen et al., 2022b). We also note that some recent advances explored more methods for visual representation, beyond convolution and attention. MLP-like models (Tolstikhin et al., 2021; Touvron et al., 2022; Hou et al., 2022; Chen et al., 2022a) directly consider a MLP layer for spatial interaction. Besides, some work employs shifting (Lian et al., 2021; Huang et al., 2021) or pooling (Yu et al., 2022c) for local communication. Similar to our work that treats the image as unordered data set, Vision GNN (ViG) (Han et al., 2022) extracts graph-level features for visual tasks. Differently, we directly apply the clustering method from conventional image processing and exhibit promising generalization ability and interpretability.

## 3   METHOD

Context Clusters forgo the fashionable convolution or attention in favor of novelly considering the classical algorithm, clustering, for the representation of visual learning. In this section, we first describe the Context Clusters pipeline. The proposed context cluster operation (as shown in Fig. 2) for feature extraction is then thoroughly explained. After that, we set

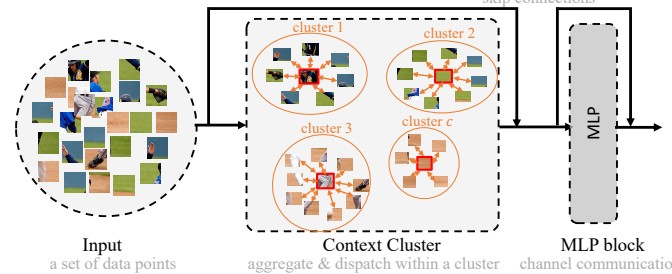

Figure 2: A Context Cluster block. We use a context cluster operation to group a set of data points, and then communicate the points within clusters. An MLP block is applied later.

up the Context Cluster architectures. Finally, some open discussions might aid individuals in comprehending our work and exploring more directions following our Context Cluster.

### 3.1   CONTEXT CLUSTERS PIPELINE

**From Image to Set of Points.**   given an input image $\mathbf{I} \in \mathbb{R}^{3 \times w \times h}$, we begin by enhancing the image with the 2D coordinates of each pixel $\mathbf{I}_{i,j}$, where each pixel's coordinate is presented as $\left[ \frac{i}{w} - 0.5, \frac{j}{h} - 0.5 \right]$. It is feasible to investigate further positional augmentation techniques to potentially improve performance. This design is taken into consideration for its simplicity and practicality. The augmented image is then converted to a collection of points (*i.e.*, pixels) $\mathbf{P} \in \mathbb{R}^{5 \times n}$, where $n = w \times h$ is the number of points, and each point contains both feature (color) and position (coordinates) information; hence, the points set could be unordered and disorganized.

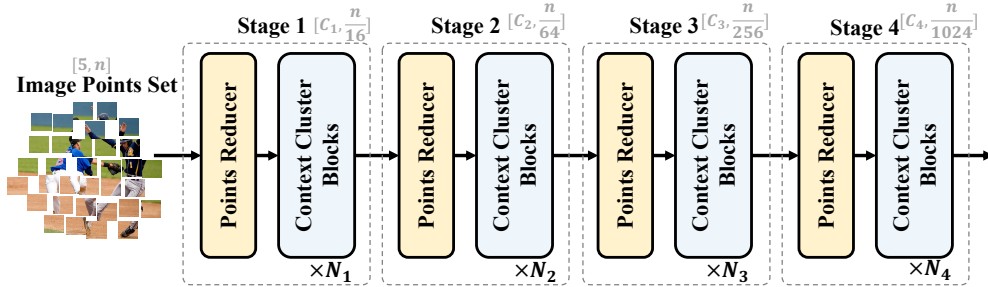

Figure 3: Context Cluster architecture with four stages. Given a set of image points, Context Cluster gradually reduces the point number and extracts deep features. Each stage begins with a points reducer, after which a succession of context cluster blocks is used to extract features.

We are rewarded with excellent generalization ability by offering a fresh perspective of image, a set of points. A set of data points can be considered as a universal data representation because data in most domains can be given as a combination of feature and position information (or either of the two). This inspires us to conceptualize an image as a set of points.

**Feature Extraction with Image Set Points.** Following the ConvNets methodology (He et al., 2016; Liu et al., 2022), we extract deep features using context cluster blocks (see Fig. 2 for reference and § 3.2 for explanation) hierarchically. Fig. 3 shows our Context Cluster architecture. Given a set of points $\mathbf{P} \in \mathbb{R}^{5 \times n}$, we first reduce the points number for computational efficiency, then a succession of context cluster blocks are applied to extract features. To reduce the points number, we evenly select some anchors in space, and the nearest $k$ points are concatenated and fused by a linear projection. Note that this reduction can be achieved by a convolutional operation if all points are arranged in order and $k$ is properly set (*i.e.*, 4 and 9), like in ViT (Dosovitskiy et al., 2020). For clarity on the centers and anchors stated previously, we strongly suggest the readers check appendix § B.

**Task-Specific Applications.** For classification, we average all points of the last block's output and use a FC layer for classification. For downstream dense prediction tasks like detection and segmentation, we need to rearrange the output points by position after each stage to satisfy the needs of most detection and segmentation heads (*e.g.*, Mask-RCNN (He et al., 2017)). In other words, Context Cluster offers remarkable flexibility in classification task, but is limited to a compromise between dense prediction tasks' requirements and our model configurations. We expect innovative detection & segmentation heads (like DETR (Carion et al., 2020)) can seamlessly integrate with our method.

## 3.2 CONTEXT CLUSTER OPERATION

In this subsection, we introduce the key contribution in our work, the context cluster operation. Holistically, we first group the feature points into clusters; then, feature points in each cluster will be aggregated and then dispatched back, as illustrated in Fig. 1.

**Context Clustering.** Given a set of feature points $\mathbf{P} \in \mathbb{R}^{n \times d}$, we group all the points into several groups based on the similarities, with each point being solely assigned to one cluster. We first linearly project $\mathbf{P}$ to $\mathbf{P}_s$ for similarity computation. Following the conventional SuperPixel method SLIC (Achanta et al., 2012), we evenly propose $c$ centers in space, and the center feature is computed by averaging its $k$ nearest points. We then calculate the pair-wise cosine similarity matrix $\mathbf{S} \in \mathbb{R}^{c \times n}$ between $\mathbf{P}_s$ and the resulting set of center points. Since each point contains both feature and position information, while computing similarity, we implicitly highlight the points' distances (locality) as well as the feature similarity. After that, we allocate each point to the most similar center, resulting in $c$ clusters. Of note is that each cluster may have a different number of points. In extreme cases, some clusters may have zero points, in which case they are redundant.

**Feature Aggregating.** We dynamically aggregate all points in a cluster based on the similarities to the center point. Assuming a cluster contains $m$ points (a subset in $\mathbf{P}$) and the similarity between the $m$ points and the center is $s \in \mathbb{R}^m$ (a subset in $\mathbf{S}$ ), we map the points to a value space to get $P_v \in \mathbb{R}^{m \times d'}$, where $d'$ is the value dimension. We also propose a center $v_c$ in the value space like the

clustering center proposal. The aggregated feature $g \in \mathbb{R}^{d'}$ is given by:

$$g = \frac{1}{C}\left(v_c + \sum_{i=1}^{m} \text{sig}\,(\alpha s_i + \beta) * v_i\right), \qquad \text{s.t.,} \quad C = 1 + \sum_{i=1}^{m} \text{sig}\,(\alpha s_i + \beta). \qquad (1)$$

Here $\alpha$ and $\beta$ are learnable scalars to scale and shift the similarity and $\text{sig}(\cdot)$ is a sigmoid function to re-scale the similarity to $(0, 1)$. $v_i$ indicates $i$-th point in $P_v$. Empirically, this strategy would achieve much better results than directly applying the original similarity because no negative value is involved. Softmax is not considered since the points do not contradict with one another. We incorporate the value center $v_c$ in Eq. 1 for numerical stability[1] as well as further emphasize the locality. To control the magnitude, the aggregated feature is normalized by a factor of $C$.

**Feature Dispatching.** The aggregated feature $g$ is then adaptively dispatched to each point in a cluster based on the similarity. By doing so, the points can communicate with one another and shares features from all points in the cluster, as shown in Fig. 1. For each point $p_i$, we update it by

$$p_i' = p_i + \text{FC}\,(\text{sig}\,(\alpha s_i + \beta) * g). \qquad (2)$$

Here we follow the same procedures to handle the similarity and apply a fully-connected (FC) layer to match the feature dimension (from value space dimension $d'$ to original dimension $d$).

**Multi-Head Computing.** We acknowledge the multi-head design in the self-attention mechanism (Vaswani et al., 2017) and use it to enhance our context cluster. We consider $h$ heads and set the dimension number of both value space $\mathbf{P}_v$ and similarity space $\mathbf{P}_s$ to $d'$ for simplicity. The outputs of multi-head operations are concatenated and fused by a FC layer. The multi-head architecture also contributes to a satisfying improvement in our context cluster, as we empirically demonstrate.

### 3.3 ARCHITECTURE INITIALIZATION

While Context Cluster is fundamentally distinct from convolution and attention, the design philosophies from ConvNets and ViTs, such as hierarchical representations and meta Transformer architecture (Yu et al., 2022c), are still applicable to Context Cluster. To align with other networks and make our method compatible with most detection and segmentation algorithms, we progressively reduce the number of points by a factor of 16, 4, 4, and 4 in each stage. We consider 16 nearest neighbors for selected anchors in the first stage, and we choose their 9 nearest neighbors in the rest stages.

An underlying issue is computational efficiency. Assume we have $n$ d-dimensional points and $c$ clusters, the time complexity to calculate the feature similarity would be $O(ncd)$, which is unacceptable when the input image resolution is high (*e.g.*, $224 \times 224$). To circumvent this problem, we introduce region partition by splitting the points into several local regions like Swin Transformer (Liu et al., 2021b), and compute similarity locally. As a result, when the number of local regions is set to $r$, we noticeably lower the time complexity by a factor of $r$, from $O(ncd)$ to $O\left(r\frac{n}{r}\frac{c}{r}d\right)$. See appendix § A for detailed configurations. Note that if we split the set of points to several local regions, we limit the receptive field for context cluster, and no communications among local regions are available.

### 3.4 DISCUSSION

**Fixed or Dynamic centers for clusters?** Both conventional clustering algorithms and SuperPixel techniques iteratively update the centers until converge. However, this will result in exorbitant computing costs when clustering is used as a key component in each building block. The inference time will increase exponentially. In Context Cluster, we view fixed centers as an alternative for inference efficiency, which can be considered as a compromise between accuracy and speed.

**Overlap or non-overlap clustering?** We allocate the points solely to a specific center, which differs from previous point cloud analysis design philosophies. We intentionally adhere to the conventional clustering approach (non-overlap clustering) since we want to demonstrate that the simple and traditional algorithm can serve as a generic backbone. Although it might produce higher performance, overlapped clustering is not essential to our approach and could result in extra computing burdens.

---

[1]If there were no $v_c$ involved and no points are grouped into the cluster coincidentally, $C$ would be zero, and the network cannot be optimized. In our research, this conundrum occurs frequently. Adding a small value like $1e^{-5}$ does not help and would lead to the problem of vanishing gradients.

Table 1: Comparison with representative backbones on ImageNet-1k benchmark. Throughput (images / s) is measured on a single V100 GPU with a batch size of 128, and is averaged by the last 500 iterations. All models are trained and tested at 224×224 resolution, except ViT-B and ViT-L.

| | Method | Param. | GFLOPs | Top-1 | Throughputs (images/s) |
|---|---|---|---|---|---|
| **MLP** | ♣ ResMLP-12 (Touvron et al., 2022) | 15.0 | 3.0 | 76.6 | 511.4 |
| | ♣ ResMLP-24 (Touvron et al., 2022) | 30.0 | 6.0 | 79.4 | 509.7 |
| | ♣ ResMLP-36 (Touvron et al., 2022) | 45.0 | 8.9 | 79.7 | 452.9 |
| | ♣ MLP-Mixer-B/16 (Tolstikhin et al., 2021) | 59.0 | 12.7 | 76.4 | 400.8 |
| | ♣ MLP-Mixer-L/16 (Tolstikhin et al., 2021) | 207.0 | 44.8 | 71.8 | 125.2 |
| | ♣ gMLP-Ti (Liu et al., 2021a) | 6.0 | 1.4 | 72.3 | 511.6 |
| | ♣ gMLP-S (Liu et al., 2021a) | 20.0 | 4.5 | 79.6 | 509.4 |
| **Attention** | ♦ ViT-B/16 (Dosovitskiy et al., 2020) | 86.0 | 55.5 | 77.9 | 292.0 |
| | ♦ ViT-L/16 (Dosovitskiy et al., 2020) | 307 | 190.7 | 76.5 | 92.8 |
| | ♦ PVT-Tiny (Wang et al., 2021) | 13.2 | 1.9 | 75.1 | - |
| | ♦ PVT-Small (Wang et al., 2021) | 24.5 | 3.8 | 79.8 | - |
| | ♦ T2T-ViT-7 (Yuan et al., 2021a) | 4.3 | 1.1 | 71.7 | - |
| | ♦ DeiT-Tiny/16 (Touvron et al., 2021) | 5.7 | 1.3 | 72.2 | 523.8 |
| | ♦ DeiT-Small/16 (Touvron et al., 2021) | 22.1 | 4.6 | 79.8 | 521.3 |
| | ♦ Swin-T (Liu et al., 2021b) | 29 | 4.5 | 81.3 | - |
| **Convolution** | ♠ ResNet18 (He et al., 2016) | 12 | 1.8 | 69.8 | 584.9 |
| | ♠ ResNet50 (He et al., 2016) | 26 | 4.1 | 79.8 | 524.8 |
| | ♠ ConvMixer-512/16 (Trockman et al., 2022) | 5.4 | - | 73.8 | - |
| | ♠ ConvMixer-1024/12 (Trockman et al., 2022) | 14.6 | - | 77.8 | - |
| | ♠ ConvMixer-768/32 (Trockman et al., 2022) | 21.1 | - | 80.16 | 142.9 |
| **Cluster** | ♥ Context-Cluster-Ti (ours) | 5.3 | 1.0 | 71.8 | 518.4 |
| | ♥ Context-Cluster-Ti‡ (ours) | 5.3 | 1.0 | 71.7 | 510.8 |
| | ♥ Context-Cluster-Small (ours) | 14.0 | 2.6 | 77.5 | 513.0 |
| | ♥ Context-Cluster-Medium (ours) | 27.9 | 5.5 | 81.0 | 325.2 |

## 4 EXPERIMENTS

We validate Context Cluster on ImageNet-1K (Deng et al., 2009), ScanObjectNN (Uy et al., 2019), MS COCO (Lin et al., 2014), and ADE20k (Zhou et al., 2017) datasets for image classification, point cloud classification, object detection, instance segmentation, and semantic segmentation tasks.

Even we are not in pursuit of state-of-the-art performance like ConvNeXt (Liu et al., 2022) and DaViT (Ding et al., 2022a), Context Cluster still presents promising results on all tasks. Detailed studies demonstrate the interpretability and the generalization ability of our Context Cluster.

### 4.1 IMAGE CLASSIFICATION ON IMAGENET-1K

We train Context Clusters on the ImageNet-1K training set (about 1.3M images) and evaluate upon the validation set. In this work, we adhere to the conventional training recipe in (Dai et al., 2021; Wightman, 2019; Touvron et al., 2021; Yu et al., 2022c). For data augmentation, we mainly adopt random horizontal flipping, random pixel erase, mixup, cutmix, and label smoothing. AdamW (Loshchilov & Hutter, 2019) is used to train all of our models across 310 epochs with a momentum of 0.9 and a weight decay of 0.05. The learning rate is set to 0.001 by default and adjusted using a cosine schedular (Loshchilov & Hutter, 2017). By default, the models are trained on 8 A100 GPUs with a 128 mini-batch size (that is 1024 in total). We use Exponential Moving Average (EMA) to enhance the training, similar to earlier studies (Guo et al., 2022b; Touvron et al., 2021). Table 1 reports the parameters used, FLOPs, classification accuracy, and throughputs. ‡ denotes a different region partition approach that we used to divide the points into [49, 49, 1, 1] in the four stages.

Empirically, results in Table 1 indicate the effectiveness of our proposed Context Cluster. Our Context Cluster is capable of attaining comparable or even better performance than the widely-used baselines using a similar number of parameters and FLOPs. With about 25M parameters, our Context Cluster surpasses the enhanced ResNet50 (Wightman et al., 2021) and PVT-small by 1.1% and achieves 80.9%

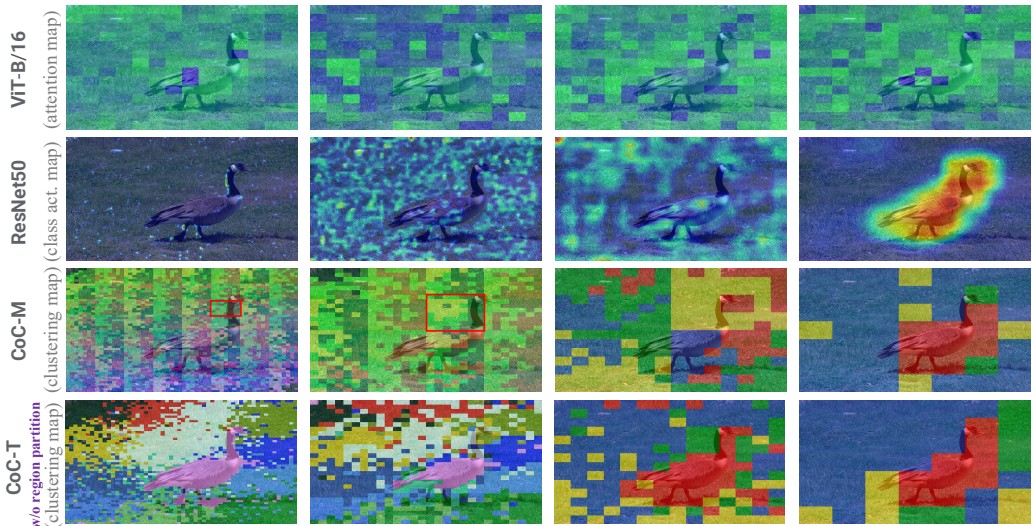

Figure 4: Visualization of activation map, class activation map, and clustering map for ViT-B/16, ResNet50, our CoC-M, and CoC-T without region partition, respectively. We plot the results of the last block in the four stages from left to right. For ViT-B/16, we select the [3rd, 6th, 9th, 12th] blocks, and show the cosine attention map for the `cls-token`. The clustering maps show that our Context Cluster is able to cluster similar contexts together, and tell what model learned visually.

top-1 accuracy. Additionally, our Context Cluster obviously outperforms MLP-based methods. This phenomenon indicates that the performance of our method is not credited to MLP blocks, and context cluster blocks substantially contribute to the visual representation. The performance differences between Context-Cluster-Ti and Context-Cluster-Ti‡ are negligible, demonstrating the robustness of our Context Cluster to the local region partitioning strategy. We recognize that our results cannot match the SOTA performance (*e.g.*, CoAtNet-0 arrives 81.6% accuracy with a comparable number of parameters as CoC-Tiny), but we emphasize that we are pursuing and proving the viability of a new feature extraction paradigm. We successfully forsake convolution and attention in our networks by conceptualizing image as a set of points and naturally applying clustering algorithm for feature extraction. In contrast to convolution and attention, our context cluster has excellent generalizability to other domain data and enjoys promising interpretability.

**Component Ablation**. Table 2 reports the results on ImageNet-1K of eliminating each individual component in Context-Cluster-Small variant. To remove the multi-head design, we utilize one head for each block and set the head dimension number to [16, 32, 96, 128] in the four stages, respectively. When the positional information is removed, the model becomes untrainable since points are disorganized. A similar phenomenon can also be seen from cifar (Krizhevsky et al., 2009) datasets. Performance dropped 3.3% without the context cluster operation. Besides, multi-head design is able to boost the result by 0.9%. Combining all the components, we arrive at a 77.5% top-1 accuracy.

Table 2: Component ablation studies of Context-Cluster-Small on ImageNet-1k.

| position info. | context cluster | multi head | top-1 acc. |
|---|---|---|---|
| ✗ | ✗ | ✗ | - |
| ✓ | ✗ | ✗ | 74.2(↓3.3) |
| ✓ | ✓ | ✗ | 76.6(↓0.9) |
| ✓ | ✓ | ✓ | **77.5** |

## 4.2 VISUALIZATION OF CLUSTERING

To better understand Context Cluster, we draw the clustering map in Fig. 4, and we also show the attention map of ViTs and the class activation map (*i.e.*, CAM) (Zhou et al., 2016) of ConvNets. Notice that the three kinds of maps are conceptually different and cannot be compared directly. We list the other two (attention and class activation) maps for reference and demonstrate the inner operations in ViTs, ConvNets, and our Context Cluster. Detail settings can be found in the caption of Fig. 4.

As the number of points is reduced, the details are merged to form a context cluster. Three observations justify the correctness and effectiveness of our Context Cluster. First, our method clearly clusters the

goose as one object context in the last stage and groups the background grass together. A similar phenomenon can also be observed from the previous stages but in more detailed and local regions. Second, our context cluster can even cluster similar contexts in the very early stages (*e.g.*, the first and second stages). Zoom in the details in the red boxes, we can see that the points belonging to the goose's neck are clearly clustered together, suggesting the strong clustering ability of our method. Last, we notice that most clusters emphasize the locality, while some (colored in bright green) show the globality a lot, as shown in the clustering map of the last stage. This further demonstrates the design philosophy; we encourage similar points to be grouped but make no restriction to the receptive field. Visual clustering map and detailed analysis indicate that our Context Cluster is effective and exhibit promising interpretability. Notably, our method demonstrates promising clustering results in a SuperPixel-style when removing the region partition operation. See appendix for more examples.

### 4.3 3D Point Cloud Classification on ScanObjectNN

Context Clusters are a natural fit for point clouds Qi et al. (2017b); Lu et al. (2022). Therefore we also examine our method for the task of point cloud classification. We choose PointMLP (Ma et al., 2022) as the foundation for our model because of its performance and ease of use. In detail, we only consider one head and set the head dimension number to $\min\left(\frac{c}{4}, 32\right)$ where $c$ indicates the channel number in each layer. We place our Context Cluster block before each Residual Point Block in PointMLP. The resulting model is termed PointMLP-CoC. Note that

Table 3: Classification results on ScanObjectNN. All results are reported on the most challenging variant (PB_T50_RS).

| Method | mAcc(%) | OA(%) |
|---|---|---|
| ♠ SpiderCNN (Xu et al., 2018) | 69.8 | 73.7 |
| ♠ DGCNN (Wang et al., 2019) | 73.6 | 78.1 |
| ♠ PointCNN (Li et al., 2018) | 75.1 | 78.5 |
| ♠ GBNet (Qiu et al., 2021) | 77.8 | 80.5 |
| ♦ PointBert (Yu et al., 2022d) | - | 83.1 |
| ♦ Point-MAE (Pang et al., 2022) | - | 85.2 |
| ♦ Point-TnT (Berg et al., 2022) | 81.0 | 83.5 |
| ♣ PointNet (Qi et al., 2017a) | 63.4 | 68.2 |
| ♣ PointNet++ (Qi et al., 2017b) | 75.4 | 77.9 |
| ♣ BGA-PN++ (Uy et al., 2019) | 77.5 | 80.2 |
| ♣ PointMLP (Ma et al., 2022) | 83.9 | 85.4 |
| ♣ PointMLP-elite (Ma et al., 2022) | 81.8 | 83.8 |
| ♥ PointMLP-CoC (ours) | **84.4**$_{\uparrow 0.5}$ | **86.2**$_{\uparrow 0.8}$ |

better settings would result in improved performance, but that is not the focus of our study. We report the mean accuracy over all classes (mAcc) and overall accuracy over all examples (OA) in Table 3.

In Table 3, we present the mean accuracy across all classes (mAcc) and the overall accuracy across all samples (OA). Experimental results show that our method can substantially increase PointMLP's performance, with improvements in mean accuracy of 0.5% (84.4% *vs.* 83.9%) and overall accuracy of 0.8% (86.2% *vs.* 85.4%). Note that the promising gain has only been made by the introduction of one head in the context cluster; with more heads and elaborate settings, performance would be improved. Most importantly, the outcomes show that our approach is highly generalizable to different domains, such as point clouds. We anticipate that our Context Cluster will operate satisfactorily when applied to more domains with little to no modifications.

### 4.4 Object Detection and Instance Segmentation on MS-COCO

Next, we investigate Context Cluster's generalisability to downstream tasks, including object detection and instance segmentation. We conduct our experiments on the MS COCO 2017 benchmark (Lin et al., 2014), which has 118k images for training and 5k images for validation. Following previous work, we train and test our model integrating with Mask RCNN (He et al., 2017) for both object detection and instance segmentation tasks. All models are trained with 1× scheduler (12 epochs) and initialized with ImageNet pre-trained weights. For comparison, we consider ResNet as a representative for ConvNets and PVT for ViTs. We report evaluation metric mean Average Precision (mAP) in Table 4.

We notice that owing to the differences in image resolution, directly adopting the Context Cluster configuration for ImageNet may not be appropriate for the downstream tasks. For classification task, we would have 49 points and 4 centers in a local region. The detection and segmentation tasks

Table 4: COCO object detection and instance segmentation results using Mask-RCNN (1×).

| Family | Backbone | Params | $AP^{box}$ | $AP^{box}_{50}$ | $AP^{box}_{75}$ | $AP^{mask}$ | $AP^{mask}_{50}$ | $AP^{mask}_{75}$ |
|---|---|---|---|---|---|---|---|---|
| Conv. | ♠ ResNet-18 | 31.2M | 34.0 | 54.0 | 36.7 | 31.2 | 51.0 | 32.7 |
| Attention | ♦ PVT-Tiny | 32.9M | 36.7 | 59.2 | 39.3 | 35.1 | 56.7 | 37.3 |
| Cluster | ♥ CoC-Small/4 | 33.6M | 35.9 | 58.3 | 38.3 | 33.8 | 55.3 | 35.8 |
| | ♥ CoC-Small/25 | 33.6M | **37.5** | **60.1** | **40.0** | **35.4** | **57.1** | **37.9** |
| | ♥ CoC-Small/49 | 33.6M | 37.2 | 59.8 | 39.7 | 34.9 | 56.7 | 37.0 |

would have 1000 points with the same configuration for image size (1280, 800). It is obvious that grouping 1000 points into 4 clusters would produce an inferior result. To this end, we investigate 4, 25, and 49 centers for a local region, and we refer to the resulting models as Small/4, Small/25, and Small/49, respectively. Results in Table 4 indicate that our Context Cluster demonstrates promising generalisability to downstream tasks. Our CoC-Small/25 outperforms the ConvNet and ViT baselines on both detection and instance segmentation tasks when properly configured (25 centers in a local region). In line with our expectations, only 4 centers cannot accurately model the large local region, and unnecessary centers cannot further enhance the performance. See appendix § C for more results.

### 4.5 SEMANTIC SEGMENTATION ON ADE20K

We examine our Context Cluster equipped with semantic FPN (Kirillov et al., 2019) for semantic segmentation task on the ADE20K (Zhou et al., 2017) dataset. For training, validation, and testing, ADE20K includes 20k, 2k, and 3k images, each of which corresponds to one of 150 semantic categories. For a fair comparison, we train all of our models for 80k iterations with a batch size of 16 on four V100 GPUs and adopt the standard data augmentation methods used in PVT (Wang et al., 2021). With an initial learning rate of 2x10-4, the AdamW optimizer is used to train all of our models. We use a polynomial decay schedule with a power of 0.9 to decrease the learning rate.

Experimental results on ADE20K are reported in Table 5. We show that our Context Clusters clearly outperform PVT and ResNet using a similar number of parameters. The promising improvements can be credited to our novel context cluster operation. Our context cluster is similar to the SuperPixel, which is an over-segmentation technology. When applied for feature extraction, we expect context cluster can over-segment the contexts in intermediate features, and show improvements for semantic segmentation tasks. Unlike in object detection and semantic segmentation tasks, the

Table 5: Semantic segmentation performance of different backbones with Semantic FPN on the ADE20K validation set.

| Backbone | Params | mIoU(%) |
|---|---|---|
| ♠ ResNet18 | 15.5M | 32.9 |
| ♦ PVT-Tiny | 17.0M | 35.7 |
| ♥ CoC-Small/4 | 17.7M | **36.6** |
| ♥ CoC-Small/25 | 17.7M | **36.4** |
| ♥ CoC-Small/49 | 17.7M | **36.3** |

centers number shows little influence on the results. More results can be found in appendix § C.

## 5 CONCLUSION

We introduce Context Cluster, a novel feature extraction paradigm for visual representation. Inspired by point cloud analysis and SuperPixel algorithms, we view an image as a set of unorganized points and employ the simplified clustering approach to extract features. In terms of image interpretation and feature extraction operation, Context Cluster is fundamentally distinct from ConvNets and ViTs, and no convolution or attention is involved in our architecture. Instead of chasing SOTA performance, we show that our Context Cluster can achieve comparable or even better results than ConvNet and ViT baselines on multiple tasks and domains. Most notably, our method shows promising interpretability and generalization properties. We hope our Context Cluster can be considered as a novel visual representation method in addition to convolution and attention.

As discussed at the end of § 3, our new perspective and design for visual representation also come with new challenges, primarily in the compromise between accuracy and speed. Better strategies are worth exploring. Departing from the current framework of detection and segmentation to apply our context cluster philosophy to other tasks is also a worthwhile direction to pursue.

## ETHICS STATEMENT

In our paper, we strictly follow the ICLR ethical research standards and laws. All datasets we employed are publicly available, and all related publications and source codes are cited appropriately.

## REPRODUCIBILITY STATEMENT

We adhere to ICLR reproducibility standards and ensure the reproducibility of our work in multiple ways, including:

- We upload our codes, pre-trained models, and the training log files to a GitHub repository, as stated in the abstract. A clear description is presented to ease the reviewing work.
- Besides the main results of each task, we also upload the codes and checkpoints for all our ablation studies to ensure each experiment is strictly proved.
- We clearly present the design of Context Cluster in Section 3.
- Detailed framework and more experiments are presented in appendix § A and § C.

By doing so, each experiment in our submission is easy to reproduce. We always open-source our research work for each submission to help the community better understand our work.

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

## A  MODEL CONFIGURATIONS

We first introduce the detailed configurations of our Context clusters in Table 6. Point reducer block is consistent with image downsize blocks, like in PVT and ConvNeXt. In the context of points, we select *k_neighbors* of the nearest points for a proposed anchor, and fuse all the points using FC layer. We reduce the number of points by a factor of *downsample_r*. The key contribution in our work is context cluster blocks. We first evenly split the whole set of points into *regions* local regions in the space. In each local region, we propose *local_centers* for clustering. We set the heads number and dimension of each head to *heads* and *head_dim*, respectively, in our context cluster operation. The channel number is expanded by a factor of *mlp_r* (*i.e.*, to $mlp\_r \times dim$) and then reduced to *dim* in the MLP block. The Context Cluster block would be repeated by several times in each stage. We mark all variables in blue for easy understanding. For the Context-Cluster-Ti‡ variation, it shares the same network structure as Context-Cluster-Ti, with the exception that we configure the region partition and local center numbers differently. In particular, the number of region partitions is set to [49, 49, 1, 1] and the number of centers in each local region is set to [16, 4, 49, 16] for the four stages.

Table 6: Detailed configurations for our Context Cluster. We initialize three variants, CoC-Tiny, CoC-Small, and CoC-Medium, with different model capacities.

| Stage | Points | Block | CoC-Tiny | CoC-Small | CoC-Medium |
|---|---|---|---|---|---|
| S1 | 50176 | Point Reducer | $k\_neighbors = 16$
$downsample\_r = 16$
$dim = 32$ | $k\_neighbors = 16$
$downsample\_r = 16$
$dim = 64$ | $k\_neighbors = 16$
$downsample\_r = 16$
$dim = 64$ |
| | 3136 | Context Cluster Blocks | $regions = 64$
$local\_centers = 4$
$heads = 4$
$head\_dim = 24$
$mlp\_r. = 8$
$dim = 32$  $\times 3$ | $regions = 64$
$local\_centers = 4$
$heads = 4$
$head\_dim = 32$
$mlp\_r. = 8$
$dim = 64$  $\times 2$ | $regions = 64$
$local\_centers = 4$
$heads = 6$
$head\_dim = 32$
$mlp\_r. = 8$
$dim = 64$  $\times 4$ |
| S2 | 3136 | Point Reducer | $k\_neighbors = 9$
$downsample\_r = 4$
$dim = 64$ | $k\_neighbors = 9$
$downsample\_r = 4$
$dim = 128$ | $k\_neighbors = 9$
$downsample\_r = 4$
$dim = 128$ |
| | 784 | Context Cluster Blocks | $regions = 16$
$local\_centers = 4$
$heads = 4$
$head\_dim = 24$
$mlp\_r. = 8$
$dim = 64$  $\times 4$ | $regions = 16$
$local\_centers = 4$
$heads = 4$
$head\_dim = 32$
$mlp\_r. = 8$
$dim = 128$  $\times 2$ | $regions = 16$
$local\_centers = 4$
$heads = 6$
$head\_dim = 32$
$mlp\_r. = 8$
$dim = 128$  $\times 4$ |
| S3 | 784 | Point Reducer | $k\_neighbors = 9$
$downsample\_r = 4$
$dim = 196$ | $k\_neighbors = 9$
$downsample\_r = 4$
$dim = 320$ | $k\_neighbors = 9$
$downsample\_r = 4$
$dim = 320$ |
| | 196 | Context Cluster Blocks | $regions = 4$
$local\_centers = 4$
$heads = 8$
$head\_dim = 24$
$mlp\_r. = 4$
$dim = 196$  $\times 5$ | $regions = 4$
$local\_centers = 4$
$heads = 8$
$head\_dim = 32$
$mlp\_r. = 4$
$dim = 320$  $\times 6$ | $regions = 4$
$local\_centers = 4$
$heads = 12$
$head\_dim = 32$
$mlp\_r. = 4$
$dim = 320$  $\times 12$ |
| S4 | 196 | Point Reducer. | $k\_neighbors = 9$
$downsample\_r = 4$
$dim = 320$ | $k\_neighbors = 9$
$downsample\_r = 4$
$dim = 512$ | $k\_neighbors = 9$
$downsample\_r = 4$
$dim = 512$ |
| | 49 | Context Cluster Blocks | $regions = 1$
$local\_centers = 4$
$heads = 8$
$head\_dim = 24$
$mlp\_r. = 4$
$dim = 320$  $\times 2$ | $regions = 1$
$local\_centers = 4$
$heads = 8$
$head\_dim = 32$
$mlp\_r. = 4$
$dim = 512$  $\times 2$ | $regions = 1$
$local\_centers = 4$
$heads = 12$
$head\_dim = 32$
$mlp\_r. = 4$
$dim = 512$  $\times 4$ |

## B  DETAIL EXPLANATIONS

One may be confused about how to specify the anchors in our point reducer block and the centers in our context cluster block. We provide illustrative and thorough explanations of them in this section.

For both anchor and center, they are generated **evenly in the space**. In order to better illustrate this, we plot **organized** image points in Fig. 5.

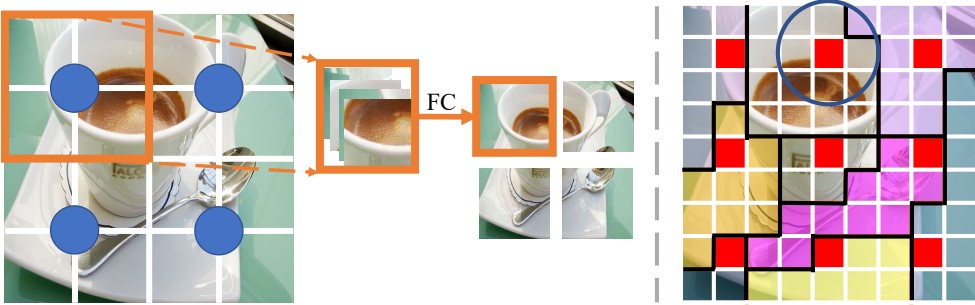

(a) Illustration of anchors for points reduction.                    (b) Demo of centers in CoC.

Figure 5: Detail explanations on the anchors in points reducer block and centers for context cluster block. The image points (represented in patch) are organized for easy understanding and illustration. In a sense, the anchors are for reducing point numbers, and centers are used for clustering. Both of them are evenly distributed in design. **On the left**, we evenly propose 4 anchors (marked in blue dot) with 4 neighbors for each anchor. **On the right**, we evenly sample 9 centers (marked in red block), hence leading to 9 irregular clusters. The center feature value is achieved by averaging its $k$ neighbors. In this figure, we show the neighbors in the big blue circle for the second center.

On the left, we display 16 points with 4 proposed anchors for point reduction, each of which takes its closest 4 neighbors into account. All neighbors are concatenated along the channel dimension, and a FC layer is used to lower the dimensional number and fuse the information. After reducing the number of points, we arrive at a new set of points with the same number of proposed anchors.

On the right, we show 9 centers (red blocks) generated from the set of image points and corresponding 9 clusters. The feature of generated centers will be given by averaging the $k$ neighbors (for the second center, we average the 9 points in the big blue circle).

The number of neighbors can be of any value. We set it to 4 or 9 due to three reasons. First, we follow the design of ConvNets and pyramid ViTs to ensure the set of points can be reorganized to a rectangular feature map. Second, this strategy eases our coding by employing convolution or pooling operations (which are equivalent to our description) and avoids heavy indices searching work. Last, a rectangular feature map is required by most detection and segmentation methods.

## C   MORE EXPERIMENTS

We conduct more experiments to validate the effectiveness of our Context Cluster.

**More results for Segmentation.** We report the CoC-Small with Semantic FPN's performance on the ADE20K validation set in Table 7 under the same conditions as previous. We notice that performance increases modestly as more centers are added in a local region, but the computational cost rises a lot. With 4 V100 GPUs, training CoC-Small/4 would take 9 hours. As a comparison, it would take 11 hours for CoC-Small/25 and 14 hours for CoC-Small/49. The cost of computing increases linearly with the complexity of the computations in our context cluster blocks.

Table 7: Semantic segmentation results of different backbones with Semantic-FPN on the ADE20K validation set.

| Family | Backbone | Params | mIoU(%) |
|---|---|---|---|
| Conv. | ♠ ResNet50 | 28.5M | 36.7 |
| Atten. | ♦ PVT-Small | 28.2M | 39.8 |
| Cluster | ♥ CoC-Medium/4 | 25.2M | **40.2** |
| Cluster | ♥ CoC-Medium/25 | 25.2M | **40.6** |
| Cluster | ♥ CoC-Medium/49 | 25.2M | **40.8** |

**More results for Detection.** We also conduct more results for object detection and instance segmentation task. Besides the experiments reported in Table 4, we also conduct experiments based on the pre-trained CoC-Medium. Results are reported in Table 8. In line with CoC-Small, CoC-Medium shows comparable performance as in ResNet50 and PVT-Small.

Table 8: COCO object detection and instance segmentation results using Mask-RCNN (1×).

| Family | Backbone | Params | $AP^{box}$ | $AP^{box}_{50}$ | $AP^{box}_{75}$ | $AP^{mask}$ | $AP^{mask}_{50}$ | $AP^{mask}_{75}$ |
|---|---|---|---|---|---|---|---|---|
| Conv. | ♠ ResNet-50 | 44.2M | 38.0 | 58.6 | 41.4 | 34.4 | 55.1 | 36.7 |
| Atten | ♦ PVT-Small | 44.1M | 40.4 | 62.9 | 43.8 | **37.8** | **60.1** | **40.3** |
| Cluster | ♥ CoC-Medium/4 | 42.1M | 38.6 | 61.1 | 41.5 | 36.1 | 58.2 | 38.0 |
| Cluster | ♥ CoC-Medium/25 | 42.1M | 40.1 | 62.8 | 43.6 | 37.4 | 59.9 | 40.0 |
| Cluster | ♥ CoC-Medium/49 | 42.1M | **40.6** | **63.3** | **43.9** | 37.6 | **60.1** | 39.9 |

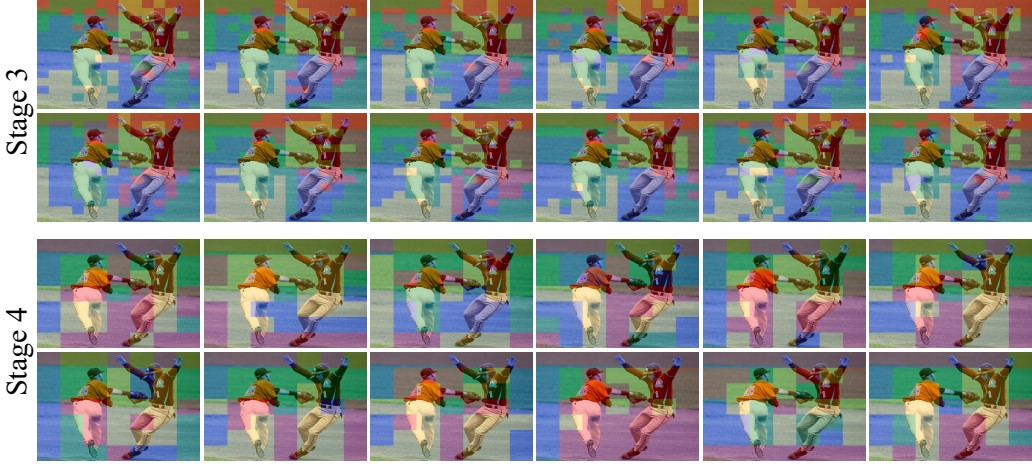

Figure 6: We visualize the clustering maps of all heads in the last block of stage3 and stage4 in our Context Cluster-Medium. While context cluster operation shows a preference for the locality as we expected, we notice that it also favors vertical or horizontal directions.

**Clustering map of all heads.** To have a better understanding of our context clustering operation, we show the clustering maps of all heads in the last block of stage3 and stage4 in our Context Cluster-Medium. As we expected, results in Fig. 6 indicate that our method is able to cluster semantically similar contexts together and exhibits decent locality. An interesting observation is that the context cluster operation also tends to cluster contexts along vertical or horizontal directions. Similar phenomena can also be observed in other images and model variants.

Table 9: Ablation study on region partition operation. Based on CoC-Tiny, we eliminated all region partition operations. To make the model trainable, the cluster numbers were changed to 16 in the first two stages and to 4 in the last two stages. We train models on 8×A100 GPUs with a batch size of 32 on each GPU, and report the training memory demand of one GPU. We test our model on one GPU.

| Model | Partition? | Parameters | FLOPs | Infer. Memory | Train Memory | Top-1 |
|---|---|---|---|---|---|---|
| CoC-Tiny | ✓ | 5.3M | 1.0G | 1.58G | 23.39G | 71.8% |
| | ✗ | 5.3M | 1.0G | 2.19G | 34.76G | 72.7% |

**Ablation on region partition operation.** While sacrificing the ability to model global interactions, region partition would introduce useful inductive biases like locality. We remove all CoC-Tiny region partition operations to see where the performance is coming from and report the results in Table 9. Experimental results indicate that without the region partition operation, the performance increased by 0.9% on ImageNet, indicating the effectiveness of our Context Cluster methodology. However, the training time and the memory demands are significantly increased (as discussed in § 3.3). Despite the fact that we agree that the region partition operation does introduce useful inductive bias, the results show that global interaction is also constructive to the success of our Context Cluster (as well as in other designs like ConvNets and ViTs).

Interestingly, we find that our Context Cluster even offers a meaningful clustering result in the early stages without the restriction of locality from region partition, as shown in Fig. 7.

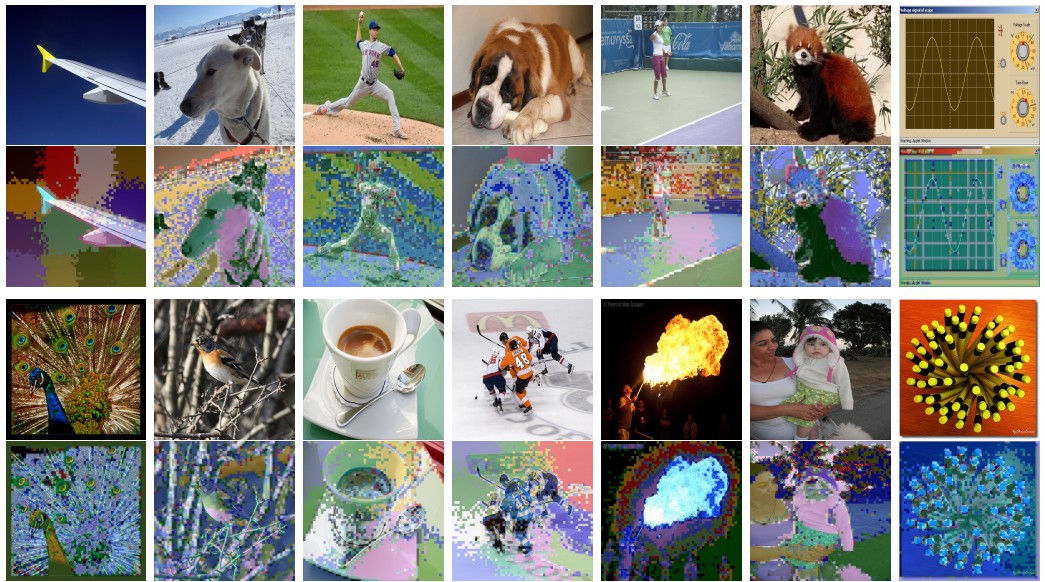

Figure 7: The clustering results of the last context cluster block in the first CoC-Tiny stage (without region partition). Without region partition, Our Context Cluster astonishingly displays "SuperPixel"-like clustering results, even in the early stage. we pick the most intriguing one out of the four heads.

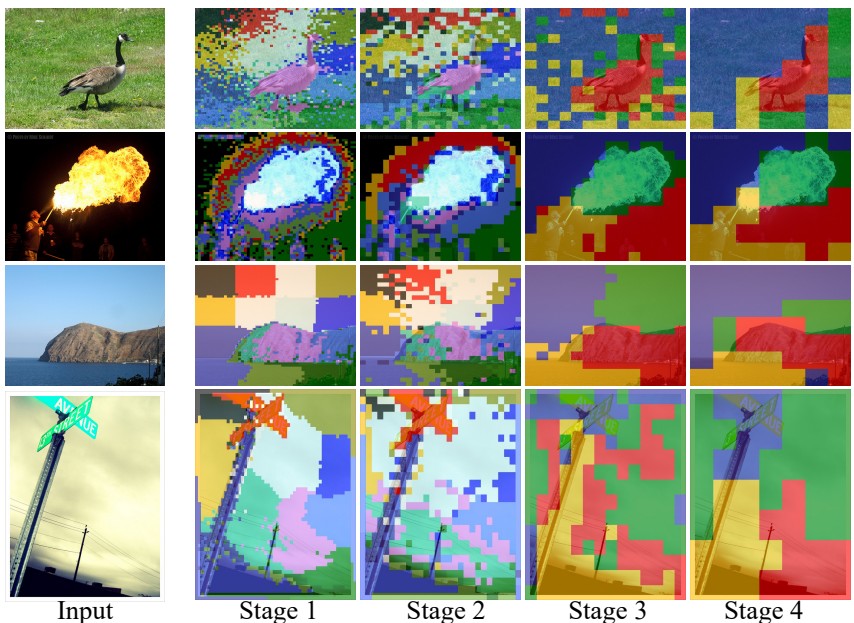

| Input | Stage 1 | Stage 2 | Stage 3 | Stage 4 |

Figure 9: Clustering results of CoC-Tiny without region partition operations.

Also, by eliminating the limitation of locality, our Context Cluster presents more promising clustering results, across all stages, as shown in Fig. 9. This phenomenon supports our motivation and indicates gratifying interpretability (which is not easy for convolution nor attention). Right figure shows an example for the clustering results of all four heads in the first stage, different clustering patterns are learned (with some noises).

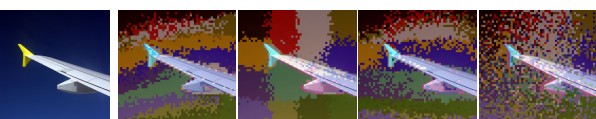

Input        Clustering results in 4 heads (16 clusters)

Figure 8: A sample of all groups' clustering results.

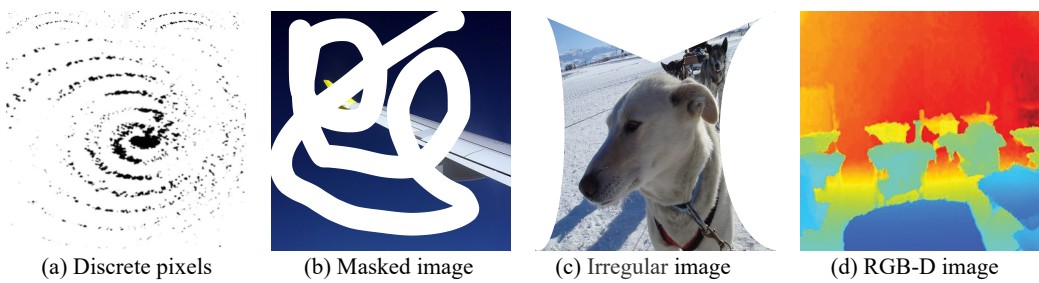

| (a) Discrete pixels | (b) Masked image | (c) Irregular image | (d) RGB-D image |

Figure 10: Four examples of image formats. Remember that there are no pixels in the white area.

In summary, these ablation studies on region partition operation indicate great potential of our Context Cluster. It would be a fascinating challenge to figure out how to eliminate the region partition operation without introducing prohibitive computation and memory requirements.

**Ablation on iteratively updating centers.** As mentioned in the discussion section, we simplified the clustering algorithm for computational efficiency by fixing the centers without updating. We also conduct a simple experiment to confirm the effectiveness of dynamic centers. By updating centers for two times, we attain a 71.85% accuracy on ImageNet-1K based on CoC-Tiny (71.83%), showing negligible improvements (0.02%).

## D    GENERALIZATION OUTLOOK

The clustering algorithm is a general method not limited to a particular input format. Previously, we validated the generalization ability of Context Cluster on both image and point clouds. Here, we further outlook the generalization ability to different image formats as shown in Fig. 10.

We starts from the discussion on the discrete pixels. Because of the clustering algorithm, our Context Cluster actually has a significant advantage over ConvNets and ViTs in processing discrete images. In other words, our Context Cluster does not require an image pixel to be in a continuous space. In detail, given an image consisting of discrete pixels, we extract features like regular images but change the center proposal method. In our submission, we describe the center proposal method by evenly proposing c centers in space, and the center feature is computed by averaging its k nearest points (which can be easily implemented by pooling). For the discrete pixels, we can consider the Farthest Point Sampling (FPS) method (Qi et al., 2017b) from point cloud processing. Notice that our method is inspired by point cloud methods, and an image with discrete pixels is naturally a point cloud set with RGB information. In addition to FPS, other discrete sampling techniques can also be investigated to propose centers for discrete pixels, including random sampling, grid sampling, *etc*.

In addition to discrete pixels, our context cluster can also be used with a variety of additional image formats. For masked images and irregular images, traditional ConvNets or ViTs require the image to be filled with white pixels. Differently, by conceptualizing an image as a set of points, we escape this step. We interpret masked or irregular images as discrete points and handle them as previously described. Thanks to the clustering algorithm, our Context Cluster exhibits great generalization ability to various image formats.

