# OpenReview forum: "Image as Set of Points"
_ICLR.cc/2023/Conference — ICLR 2023 notable top 5%_

### Official Review · Reviewer_1Gqf · 2022-10-23

**Confidence:** 4
**Correctness:** 4
**Technical Novelty And Significance:** 4
**Empirical Novelty And Significance:** 4
**Recommendation:** 10

**Clarity, Quality, Novelty And Reproducibility:**

This paper proposes a interesting idea and the architecture is novel for representation of image. The writing is good with high-quality figures and tables. The code is provided for reproducibility.

**Strength And Weaknesses:**

Strength:
+ The idea is very interesting and this new architecture could be an alternative architecture for CNN, ViT. It provides a new perspective on how the information of an image is represented.
+ The idea should be of great interest for the representation learning and computer vision community.
+ The computation is a hard problem if using clustering on 224x224 image, while the authors use a smart way to address it, similar to Swin Transformer.
+ The performance is impressive as compared with recent popular architectures.
+ The propose method also generalizes well on downstream tasks like object detection and segmentation.
+ The paper is well organized and the writing is good.

Weakness:
- The paper claims to bridge the gap between representation of image and point cloud. It seems to work well on PointNet. But it is kind of hard to understand. The authors use fixed centers (Fig.5) to save computation, how does this work if the discrete pixels are sparsely sampled from one image? It might be helpful if the authors can provide more explanation about this. It would also be interesting to see the performance of dynamic centers? For example, maybe use only 2 iteration for center updates?
- The proposed method may have good interpretation ability, but not better than CNN or ViT. The figure 4 might need more explanation since the framework is very different from other architectures. The method seems to cluster similar contexts in early stage, but it is not easy for reader to get this insight from figure 4.

**Summary Of The Paper:**

This paper proposes a different formulation for vision tasks, i.e. considering image as a set of unorganized points instead of organized rectangular shape or patches sequence. Each point has raw feature and positional information and they are grouped with clustering algorithm. It shows comparable or sometimes better result as CNN or transformer. The code is also released.

**Summary Of The Review:**

To summary, I think this paper is very interesting and should be discussed at the main conference. It might be a good alternative architecture for vision tasks. The results are impressive and well-presented. I recommend acceptance.

---

> ### Author Response · Authors · 2022-11-18
> **Thank you! Response to Reviewer 1Gqf**
>
> Dear Reviewer 1Gqf,
>
> Thank you very much for your strong support and the detailed comments.  We address your concerns as follows.
>
> --------
>
> **Question 1: Hard to understand the application on point cloud classification.**
>
> **Response:** We appreciate your kind reminder. Our context cluster is a natural fit for point cloud tasks. Unlike images, point clouds are collections of points that are represented by 3D coordinates, such as [x, y, z]. In conventional point cloud methods like PointNet++ and PointMLP, points would be clustered into local groups based on the coordinates (exactly the same as our region partition). We simply apply our context cluster operation to each group in the point cloud to enhance the performance. Empirical results show that our Context Cluster improves the performance a lot on the ScanObjectNN benchmark based on the PointMLP method.
>
>
> --------
>
>
> **Question 2: How can our method be applied to discrete pixels?**
>
> **Response:** We really appreciate this question, and we are quite interested in talking about it more. `Because of the clustering strategy, our Context Cluster actually has a significant advantage over ConvNets and ViTs in processing discrete images. In other words, our Context Cluster does not require an image pixel to be in a continuous space`.
>
> In detail, given an image consisting of discrete pixels, we extract features like regular images but change the center proposal method. In our submission, we describe the center proposal method by evenly proposing c centers in space, and the center feature is computed by averaging its k nearest points (which can be easily implemented by pooling). `For the discrete pixels, we can consider the Farthest Point Sampling (FPS) method [R1] from point cloud processing`. Notice that our method is inspired by point cloud methods, and an image with discrete pixels is naturally a point cloud set with RGB information.
>
> `In addition to FPS, other discrete sampling techniques can also be investigated to propose centers for discrete pixels`, including random sampling, grid sampling, and others.
>
> `In addition to discrete images, our context cluster can be used with a variety of additional image formats`, including RGB-D images (an image format in 3D space), masked images (a special case of discrete images), irregular images (a special case of discrete images), etc. Thanks to the clustering algorithm, our Context Cluster exhibits great generalization ability to various image formats.
>
>
> Thanks for the reviewer's question, and we will add these responses to the appendix.
>
> [R1] Qi, Charles Ruizhongtai, et al. "Pointnet++: Deep hierarchical feature learning on point sets in a metric space." NeurIPS. 2017.
>
> --------
>
>
> **Question 3: Dynamic centers like only 2 iterations for center updates?**
>
> **Response:** Thank you for the suggestion. We have conducted the required experiment, and report the results in the following table.
>
> | Method   | Update centers?    | ImageNet-1K Top-1 |
> |----------|--------------------|-------------------|
> | CoC-Tiny | No                 | 71.83     |
> | CoC-Tiny | Yes (2 iterations) | 71.85      |
>
> Empirically, we didn’t observe a significant performance gap between fixed or dynamic centers. Two possible reasons: 1) the mean feature and fixed center already provide a good representation; 2) 2 iterations may not be enough for a meaningful center updating.
>
> Instead of  iteratively updating the centers to achieve dynamic centers, a possible solution is to directly predict an offset for each center using a small network (e.g., two FC layers), like GCN and DAN. This strategy not only achieves dynamic centers, but also avoids heavy computations. In our research plan, we will investigate the feasibility  in our next version of CoC.
>
> -----------
>
> **Question 4: Better visualization and understanding on Fig. 4.**
>
> **Response:** We thank the reviewer for pointing out this issue. We will reproduce the figure and update it in the reversion for better understanding. We will keep you updated when we post new figures.
>
> ---------
>
> **In the end, thanks a lot for your detailed comments and thank you for helping us improve our work! We appreciate your thoughts on our work and we would be more than happy to discuss more during or after the rebuttal to explore more in this direction. Please let us know if you have any further questions. We are actively available until the end of this rebuttal period.**

---

> > ### Comment · Reviewer_1Gqf · 2022-11-18
> > **Response to the Authors**
> >
> > I really appreciate the new results and explanation provided in the rebuttal and my concerns are mostly addressed. I think this new version is further improved with new results, and I am raising my rating to strong accept.

---

> > > ### Author Response · Authors · 2022-11-20
> > > **Thank you for your support and constructive comments**
> > >
> > > Dear Reviewer 1Gqf,
> > >
> > > Thank you very much for your strong support and constructive comments!
> > >
> > > We updated the figures (Fig.4 & Fig.6), and presented some new results (visual and quantified) in Appendix Sec. C. We also added above discussions to Appendix Sec. D.
> > >
> > > Feel free to discuss more during or after the rebuttal phase if you have any further questions. We are actively available. Thanks again.

---

### Official Review · Reviewer_QuSd · 2022-10-24

**Confidence:** 3
**Correctness:** 4
**Technical Novelty And Significance:** 4
**Empirical Novelty And Significance:** 4
**Recommendation:** 8

**Clarity, Quality, Novelty And Reproducibility:**

The data and code are cited - so the work is technically reproducible but would not be immediately straightforward I guess?

The clarity of the methodology could be improved for ease of following.

**Strength And Weaknesses:**

The paper is well written (although requires an editorial read)  and the experiments provided are detailed.

The methodology could be more clearly presented though. It should be immediately understandable but the reader has to re-read to follow properly. As a case, I would like to be explicitly told how the context clusters are believed to capture the content better.

The citations are incorrectly used throughout.
The use of italics and bold in paragraphs seems unprofessional and I would prefer it not used.

Why are some figures bold in Table 1? It is not clear what this means.

The authors state "See appendix for more experiments." It would read well if a short explanation of what is in the appendix is provided here.

**Summary Of The Paper:**

The authors present an alternative approach to image content representation called context clusters. These are compared as alternatives to convolutional networks and vision transformer approaches.

The work is well presented and well written. They argue for their approach in a scientific manner and set it in place for future use in other vision techniques and applications.

**Summary Of The Review:**

I enjoyed this paper. The authors write with enthusiasm and back up their claim that context clusters are worth further investigation through scientific means.

---

> ### Author Response · Authors · 2022-11-18
> **Thank you! Response to Reviewer QuSd**
>
> Dear Reviewer QuSd,
>
>
> We sincerely appreciate your detailed suggestions and encouragements, such as "write with enthusiasm." We took all your suggestions into consideration and updated our manuscript.
>
> ------------
>
> **1. Clearly present the methodology.**
>
> Thanks for your kind suggestions. In order to improve readability, we have revised several details of our presentations and will continue to update our manuscript.
>
> In addition, we acknowledge that our Context Cluster is not theoretically superior to convolution and attention. We highlight that the key contribution of our study is the introduction of a new, general, interpretable paradigm for feature extraction in computer vision called context clustering. The efficacy and effectiveness of our approach are also supported by empirical findings.
>
> ------------
>
> **2. Citations are incorrectly used throughout.**
>
> We much appreciate this point, and we changed the paper's citation style. This advice greatly enhances the quality of our paper.
>
> ------------
>
> **3. Numbers bold in Table 1 and unprofessional use of italics and bold.**
>
> We appreciate the reviewer's attention to this, and to improve the clarity of our paper, we have removed all unnecessary formats.
>
> ------------
>
> **4. Detail the appendix.**
>
> For each of the statements like "See appendix for more experiments," we have given a brief explanation and a section reference.
>
> ------------
>
> **Again, thanks a lot for your detailed comments and thank you for helping us improve our work! Please let us know if you have any further questions. We are actively available until the end of this rebuttal period.**

---

### Official Review · Reviewer_Mp7d · 2022-10-25

**Confidence:** 4
**Correctness:** 3
**Technical Novelty And Significance:** 4
**Empirical Novelty And Significance:** 4
**Recommendation:** 8

**Clarity, Quality, Novelty And Reproducibility:**

- The paper is well-written and easy to follow. The authors also provide additional explanations of some model designs in the appendix which are much appreciated.
- Both the topic and the proposed method are both original.
- The general architecture is reproducible based on the model description, additional hyper-parameters are required to reproduce the experimental results.

**Strength And Weaknesses:**

Strengths:

- To the best of the reviewer’s knowledge, the topic of considering an image as a set of points and extracting features from it for vision tasks is original and very interesting.
- The proposed method that uses the clustering algorithm as the basic build block is novel and of significance to the community.
- The evaluation plan of the paper is comprehensive. It provides experiments on standard vision tasks like image classification and object detection/segmentation and applications for point cloud inputs like object classification.
- The evaluation results show that the method provides improvements on various tasks over the CNN and ViT baselines  (though not outperforming the state-of-the-art approach).

Weaknesses:

- By using the region partition mechanism, the set of points is no longer unorganized but becomes structured based on their locality. Additional experiments are required to clarify the role of the region partition.
    - Before applying the context clusters operation, the region partition operation, which is similar to the shifting windows in Swin [1], is introduced to reduce the computational cost. The authors seem to imply that the region partition trades off performance for speed. However, the locality introduced by the region partition could also bring useful inductive bias for the encoder. Therefore, additional experiments are required to answer the following questions:
        1. If the region partition operation is removed in the clustering process, could the model achieve similar or better performance? What would the clustering map be like in this case?
        2. It would be nice to introduce Swin as one baseline to investigate this problem.

[1] Liu, Z., Lin, Y., Cao, Y., Hu, H., Wei, Y., Zhang, Z., Lin, S., & Guo, B. (2021). Swin Transformer: Hierarchical Vision Transformer using Shifted Windows. *arXiv*. https://doi.org/10.48550/arXiv.2103.14030

**Summary Of The Paper:**

This paper proposes a new view of images that considers each image as a set of points (the pixels) and uses a clustering algorithm to extract the features from it. The goal is to investigate the way to utilize this new form of visual representations and evaluation the performance that could be achieved. To this end, the paper introduces a novel backbone network that includes the proposed Context Clusters and evaluation this model on several vision tasks as well as a point cloud data application.

**Summary Of The Review:**

This paper introduces a new form of image representation that considers each image as a set of points and proposes a clustering-based architecture for feature extraction. Both the idea of “image as set of points” and the proposed architecture are interesting and novel. The experiment result also shows that the method achieves comparable performance to ConvNets and ViTs. A small concern is that the role of the region partition mechanism is unclear since good performance could actually be attributed to this design.

---

## Post-rebuttal updates:
I appreciate the detailed feedback and the extra experiments. It is very interesting to see that:

1. Removing the partition operation actually improves the performance. This demonstrates the efficiency of the original design.
2. The clustering behavior reveals superpixel-based grouping. This is anticipated, but demonstrating that it is actually working in this way in the paper is much appreciated.
3. Slightly worse than Swin in performance. The answers to my questions and the performance difference are also acceptable for a novel model design.

Since most of my concerns have been addressed, I decided to raise the score to 8. This is a good paper and worth acceptance

---

> ### Author Response · Authors · 2022-11-17
> **Thank you! Responses to Reviewer Mp7d (Part 1)**
>
> Dear Reviewer Mp7d,
>
> Many thanks to your comments and your questions, which help us a lot to improve our work. We address your questions as follows.
>
> **Questions: What if the region partition operation is removed in the clustering process?**
>
> **Response:** Thanks for the question, and the local and global interaction issue is also posted by Reviewer cH7t. In our rebuttal, we carried out the required experiments to investigate the impact of region partition operation in our CoC.
>
> We remove the region partition operation in all stages. The cluster numbers in the first two stages were set to 16 and the final two stages were set to 4 in order to train the model (considering the point number in each stage). Notice that 16 is the maximum value that our GPUs can support during training.  The results are presented in the Table.
>
> | Method                               | With partition? | Parameters | FLOPs | Top-1 | Inference Memory | Train Memory |
> |--------------------------------------|-----------------|------------|-------|-------|------------------|--------------|
> | CoC-tiny                             | Yes             | 5.3        | 1.0   | 71.8  | 1.58G            | 23.39G       |
> | CoC-tiny   | No              | 5.3        | 1.0   | 72.7  | 2.19G            | 34.76G       |
>
> Experimental results indicate that without the region partition operation, the performance increased by 0.9% on ImageNet.  However, the training time and the memory demands are significantly increased.  Despite the fact that we agree that the region partition operation does introduce useful inductive bias, the results show that global interaction is also constructive to the success of our Context Cluster (as well as in other designs like ConvNets and ViTs).
>
> **More results (clustering map) will be posted later, thanks for your patience.**
>
>
> -------------------------
> **Questions: Swin Transformer baseline**
>
> **Response:** We also provide the comparison to Swin Transformer, as shown in the following table.
>
> | Method    | Parameters | FLOPs | Top-1 |
> |-----------|------------|-------|-------|
> | CoC-M     | 27.9       | 5.5   | 81.0  |
> | Swin-Tiny | 29.0       | 4.5   | 81.3  |
>
> Compared to the Swin Transformer, our method performs comparable or slightly worse. As clearly presented in our submission, the purpose of CoC is to prove concepts instead of chasing SOTA performance.  In our plan, a more powerful version of CoC (with dynamic centers, overlapped clustering, better positional embedding, global interaction, and more detailed designs) will be presented, and we believe that the enhanced CoC would perform much better than Swin and achieve promising results (comparable with more SOTA methods as our target).
>
> -----------------------------
> **Again, thank you so much for helping us improve the paper! Please let us know if you have any further questions. We are actively available until the end of this rebuttal period. Looking forward to hearing back from you!**

---

> > ### Author Response · Authors · 2022-11-20
> > **Thank you! Updates about the region partition ablation study and our new findings.**
> >
> > Dear Reviewer Mp7d,
> >
> > Thanks for your patience. We really appreciate the suggestion of removing region partition operation to verify the effectiveness of our Context Cluster.  In summary, we have two findings: **1) removing region partition operation would further improve performance (but with heavier computational cost)**; **2) a CoC model without region partition operation shows promising  clustering results, revealing excellent interpretability**.
> >
> > For 1), in the last post, we have shown that a Context Cluster model without region partition would further improve the performance (as well as introduce unaffordable computations and), indicating the performance is due to our Context Cluster rather than the region partition operation (and related inductive bias like locality).
> >
> > For 2), we found that CoC without region partition would exhibit more promising clustering results than the one with region partition, as shown in Fig.7, Fig.8, and Fig.9.  We emphasize that such superior interpretability is not easy for ConvNets and ViTs by directly visualizing the feature maps like ours.
> >
> > In summary, these ablation studies on region partition operation indicate great potential of our Context Cluster. It would be a fascinating challenge to figure out how to eliminate the region partition operation without introducing prohibitive computation and memory requirements.
> >
> > -------------
> >
> > **We have updated previous experimental results and new figures in the appendix Section C. Again, thanks a lot for helping us improve our work and inspiring us to explore more. Many thanks to your comments, and an efficient global-interacted CoC will be included in our next research plan.**

---

> ### Author Response · Authors · 2022-12-07
> **Thank you and updates about more visualization and new findings**
>
> Dear Reviewer Mp7d,
>
> Thanks a lot for raising the score and we really appreciate your questions about the ablation of region partition and more visualizations. Here, we add more analysis and more visualizations.
>
> ----
>
> 1) `CoC without region partition can achieve better "superpixel"-like clustering results at all stages, verifying our motivation and providing gratifying interpretability`. For your convenience, we randomly selected several examples and showed the clustering results here: https://anonymous.4open.science/r/ContextCluster/uploads/rebuttal_examples.png.
>
> ----
>
> 2) `In addition to context-aware clustering, our CoC can consistently learn fixed and context-irrelevant clustering patterns, such as border clustering`. We constantly observe the same phenomena across all CoC versions, including different model sizes and with or without region partition. The results are presented here:
> https://anonymous.4open.science/r/ContextCluster/uploads/rebuttal_border_cluster_stage4.png
>
> ----
>
> 3) `By removing the region partition operation in CoC, we achieve better border clustering results even in stage 3, instead of the last stage`. This result further demonstrates how our CoC can benefit from global modeling (which has already been verified quantitatively and qualitatively). Here are the improved border clustering results:
> https://anonymous.4open.science/r/ContextCluster/uploads/rebuttal_border_cluster_stage3.png
>
> ----
>
> Again, thank you for your insightful comments that greatly improve our work. The above experiments and findings will be included in our revision. For our Context Cluster, we believe that there are more potentials to be explored. If you have any further questions or concerns, please let us know. We are actively avaliable during rebuttal.

---

### Official Review · Reviewer_cH7t · 2022-10-25

**Confidence:** 4
**Correctness:** 4
**Technical Novelty And Significance:** 3
**Empirical Novelty And Significance:** 4
**Recommendation:** 8

**Clarity, Quality, Novelty And Reproducibility:**

The paper is very clearly written and of high quality. The authors are very thorough in providing reproduction details (source code, model checkpoints, etc.).
I think the paper is highly innovative. Its idea of point cloud modeling on 2D images to instantiate generic vision models is groundbreaking. It has a high future value for our field moreover. See "contribution" and "strength" parts above.

**Strength And Weaknesses:**

**[strength]**

- This is a very insightful work towards general visual backbone design. The fusion of SuperPixel idea as well as PointNet structure is natural and elegant. I believe this work reveals the potential of point cloud modeling for 2D image processing, which is certainly a great contribution to the community and is worth exploring.
- I would say that CoC is modeled in a way that fits well with the nature of visual signal processing, as it has the inductive biases that are as useful as those of convolutional networks (such as locality and multi-scale).
- The interpretability is also as good as convolution (CoC embodies spatial selectivity, while convolution exemplifies the idea of pattern matching). Besides, CoC acts like the deformable convolution and is clearly more flexible and general than ordinary convolution that operates on regular grids.


**[weakness]**
- **Potential representation weakness**. As an architecture similar to ConvNet and PointNet, CoC could be significantly weaker than Transformers in terms of global modeling capabilities (e.g., interacting two spatially distant objects). What are the authors' insights on this problem?
- **Related work**. I find the first paragraph in Related Work is way more relevant to CoC than the latter two. It would be much appreciated if the authors could devote more space to the first paragraph, e.g., introduce [R1,R2,R3,R4] in more detail and discuss them with this paper in terms of method philosophy.
- **Figure clarity**. I recommend using more contrasting colors when visualizing. Currently these two figures (Fig. 4 and 6) do not illustrate the clustering effect of CoC models very well.
- **Font formats**. The spacing between characters in the headings is too small, making them look cramped. Also, there seems to be too much use of bolding or italics in the body text. I would suggest using a more concise format.

------------

**[open questions]**

I have more questions as follows and would like to discuss them further with the authors:

- From Fig. 5(b), the CoC block has similarities with DAT [R5]. Can the authors further analyze or compare them?
- I don't think it's necessarily a disadvantage that the methods [R3, R4] can only be used for specific tasks like segmentation; in other words, a point-based (or SuperPixel-based) modeling might be inherently more useful for dense prediction tasks (detection, segmentation, etc.) than classification. The authors mentioned that they expected to see more work that seamlessly integrates CoC and DETR in the future. This is reasonable, but I still expect the authors to come up with some prototypes of CoC for dense prediction :).

----------
[R1] Varun Jampani, Deqing Sun, Ming-Yu Liu, Ming-Hsuan Yang, and Jan Kautz. Superpixel sampling networks. In ECCV, 2018.

[R2] Zixuan Huang and Yin Li. Interpretable and accurate fine-grained recognition via region grouping. In CVPR, 2020.

[R3] Fengting Yang, Qian Sun, Hailin Jin, and Zihan Zhou. Superpixel segmentation with fully convolutional networks. In CVPR, 2020.

[R4] Qihang Yu, Huiyu Wang, Dahun Kim, Siyuan Qiao, Maxwell Collins, Yukun Zhu, Hartwig Adam, Alan Yuille, and Liang-Chieh Chen. Cmt-deeplab: Clustering mask transformers for panoptic segmentation. In CVPR, 2022a.

[R5] Xia, Zhuofan, et al. Vision transformer with deformable attention. In CVPR, 2022.


**Summary Of The Paper:**

This is a pioneering work considering SuperPixel-like idea and point cloud modeling for 2D general visual backbone.
The main contributions are:
- Proposing a model called CoC to explore the possibility of point-cloud-clustering modeling for 2D image processing, and initially verifiing its effectiveness.
- Providing fresh insights into the design of general vision model, which are not only well suited to cope with dense prediction tasks (detection and segmentation), but also facilitate the combination with 3D point cloud data in order to promise a very general cross-modal vision model.

**Summary Of The Review:**

This work is innovative, insightful, written with ample detail, and highly reproducible.
I believe it deserves to be published and discussed in ICLR.

---

> ### Author Response · Authors · 2022-11-17
> **Thank you! Responses to Reviewer cH7t (Part 1)**
>
> Dear Reviewer cH7t,
>
> Thank you very much for your support and constructive suggestions.  We address your concerns as follows.
>
> **Question 1. Potential representation weakness on global modeling capabilities ( e.g., interacting two spatially distant objects).**
>
> **Response:** Local and global interactions are popular study topics in computer vision and other communities. Our CoCs are initially designed to be globally aware. Due to the computational limitations, we introduce the region partition strategy to reduce the computational complexity. Here, we provide four perspectives in response to this question (global interaction).
>
>
> 1) The region partitioning strategy is optional for CoCs and can be removed for better global interaction modeling. (as pointed out by reviewer Mp7d). However, removing region partitioning  would result in time-consuming computations, heavy memory costs, longer training time, etc., as stated in Section 3.3. As suggested, our experiments showed that though the computation is heavy, this approach still has its merits, and even achieves better performance by introducing more global interaction.
>
>
> | Method                               | With partition? | Parameters | FLOPs | Top-1 | Inference Memory | Train Memory |
> |--------------------------------------|-----------------|------------|-------|-------|------------------|--------------|
> | CoC-tiny                             | Yes             | 5.3        | 1.0   | 71.8  | 1.58G            | 23.39G       |
> | CoC-tiny   | No              | 5.3        | 1.0   | 72.7  | 2.19G            | 34.76G       |
>
>
> 2) A traditional ViT-style isotropic design is an option. The number of points can be drastically reduced by a factor of 256 (considering 256 neighbor points instead of 16 neighbor points) in the first Points Reducer, yielding 196 points if the input size is 224x224. The computational demands would be greatly decreased with this method. Then, instead of using an area partitioning approach, isotropic CoC blocks can be used to extract deep global features  like typical ViTs.
>
>
>
>
> 3) By creating interactions among the cluster centers in different partitions, we can implicitly capture global interaction. To build the interactions, we can introduce another clustering, or other aggregation methods like pooling operations.
>
> 4) We can take advantage of a variety of earlier techniques, such as LambdaNet [R6], SENet [R7], and GENet [R8], to introduce global interaction.
>
> The methods outlined above may be used without much modification to generate global interactions and can be applied immediately to our CoC. We look forward to exploring them in our future work.
> Meanwhile, we think it's also intriguing and worthwhile to study or revisit some techniques that derive from clustering algorithms to create global interaction without requiring intensive computation.
>
> [R6] Bello, Irwan. "Lambdanetworks: Modeling long-range interactions without attention." ICLR. 2021.
>
> [R7] Hu, Jie, Li Shen, and Gang Sun. "Squeeze-and-excitation networks." CVPR. 2018.
>
> [R8] Hu, Jie, et al. "Gather-excite: Exploiting feature context in convolutional neural networks." NeurIPS, 2018.
>
> ---------------------
>
> **Question 2.  Related Work.**
>
> **Response:** Thank you for your thoughtful advice. In the related work, we go into detail about these methods, and we update them in our revised version.
>
> ---------------------
>
> **Question 3. Figure Clarity.**
>
> **Response:** We appreciate the suggestion. We will update the figures in our revision soon.
>
> ---------------------
>
> **Question 4. Font Format.**
>
> **Response:** We also updated the font format in our revised version to make our manuscript clear.

---

> > ### Author Response · Authors · 2022-11-17
> > **Thank you! Responses to Reviewer cH7t (Part 2)**
> >
> > **Open Question 1: About the DAT (Deformable Attention Transformer).**
> >
> > **Response:**
> > We appreciate you bringing this insightful work to our notice, and we will cite this work in our revision.
> >
> > It's interesting that Fig. 5(b) in our CoC submission and Fig. 1(d) in DAT look similar, but they are essentially different.
> >
> > In conclusion, DAT [R5] brings the DCN [R9] methodology to Vision Transformers in order to increase computational efficiency, focusing on tokens relevant to queries; While our CoC takes fixed centers and non-overlapped clustering into consideration for feature extraction.
> >
> > Meanwhile, in terms of learnable offsets, DAT and CoC can be related. As opposed to k-means clustering approaches, our CoCs' centers are evenly initialized and do not undergo iterative updating (also pointed out by reviewer 1Gqf). Iterative updating was not taken into consideration because it would greatly increase the computation and time for inferences (as well as training). To tackle this issue, we can learn an offset for each center, like DAT and DCN.
> >
> > We value the reviewer's guidance in introducing this work and helping us in polishing our work.
> >
> >
> > ----------------------
> > **Open Question 2: Come up with some prototypes of CoC for dense prediction.**
> >
> > **Response:** More prototypes based on the concept of CoC are worthwhile to investigate, as we stated in the conclusion section. We are aware, though, that this is a large and promising prospect. Here, we present some preliminary insights.
> >
> > 1. We believe Context Cluster can even be a natural substitute of self-attention in DETR [R10] and Deformable DETR [R11].
> >
> > 2. We embrace the concept from MaskFromer [R12] and Mask2Former [R13] that pre-pixel classification is not the only option for segmentation problems, such as semantic segmentation and instance segmentation. In addition to pre-pixel classification and pre-mask classification, pre-cluster classification is another option that can be done more succinctly.
> >
> > 3. For detection tasks, we notice that some work already considers points as objects, like CornerNet [R14] and CenterNet [R15]. We believe our Context Cluster could also be applied to detection tasks in a similar way.
> >
> > There would be a lot of potential work and directions  to come up with some prototypes of CoC for dense prediction, and we have great interests in new prototypes for these tasks.
> >
> > [R9] Dai, Jifeng, et al. "Deformable convolutional networks." ICCV. 2017.
> >
> > [R10] Carion, Nicolas, et al. "End-to-end object detection with transformers." ECCV, 2020.
> >
> > [R11] Zhu, Xizhou, et al. "Deformable detr: Deformable transformers for end-to-end object detection." ICLR. 2021.
> >
> > [R12] Cheng, Bowen, Alex Schwing, and Alexander Kirillov. "Per-pixel classification is not all you need for semantic segmentation." NeurIPS. 2021.
> >
> > [R13] Cheng, Bowen, et al. "Masked-attention mask transformer for universal image segmentation." CVPR. 2022.
> >
> > [R14] Law, Hei, and Jia Deng. "Cornernet: Detecting objects as paired keypoints." ECCV. 2018.
> >
> > [R15] Zhou, Xingyi, Dequan Wang, and Philipp Krähenbühl. "Objects as points." arXiv. 2019.
> >
> > -----------------------------------
> >
> > **Lastly, thank you so much for helping us improve the paper and appreciate your open discussions! Please let us know if you have any further questions. We are actively available until the end of this rebuttal period. Looking forward to hearing back from you!**

---

> > > ### Author Response · Authors · 2022-12-04
> > > **Updates: another simple way to introduce global modeling capabilities and improve the performance**
> > >
> > > Dear Reviewer cH7t,
> > >
> > > For the **potential representation weakness** issue, we also performed additional experiments.
> > >
> > > Besides directly removing the region partition operation, we explored another simple approach to add global modeling capabilities to our Context Cluster: we directly extracted the global context via global average pooling and added the pooled features to the input. That is, for each Context Cluster block, we update the input $x$ by $x' = x + \alpha*GAP(x)$, where $\alpha \in R^c$ is a learnable vector and $GAP$ stands for global average pooling. This simple operation introduces almost no additional computations and parameters but improves the performance of CoC-tiny by **0.6% (from 71.8% to 72.4%)**.
> > >
> > > `During our rebuttal, we tried two straightforward and simple solutions to introduce global modeling capacities to our Context Cluster and validated the effectiveness. We believe that many approaches, as we previously suggested, would also enable global modeling and enhance empirical performance for our Context Cluster. It would also be interesting to explore more proper methods for global modeling.`
> > >
> > > Thanks for your comments and suggestions, we will add these experiments to our revision. Feel free to let us know if you have any further questions or concerns.

---

> ### Author Response · Authors · 2022-12-11
> **Thank you! Updates and highlights.**
>
> Dear Reviewer cH7t,
>
> Again, we would like to express our sincere thanks for your insightful comments and open discussions, which have greatly improved our work and inspired us to research more!
>
>
> **We hope our replies can address your concerns and that our responses to the open discussions can convince you. If you have any additional questions or anything you would like to discuss in more detail, please feel free to let us know (the Author/Reviewer/AC discussion deadline of 12/12 is quickly approaching). We would be more than happy to discuss further and respond promptly.**
>
> ----
>
> Here, we highlight and summarize some noteworthy points made during our rebuttal:
>
> 1. Introducing global modeling capabilities consistently enhances empirical performance, as we posted previously.
>
> 2. Our Context Cluster is able to exhibit superpixel-like clustering results, as shown here: https://anonymous.4open.science/r/ContextCluster/uploads/rebuttal_examples.png
>
> 3. Besides the concise design philosophy, our Context Cluster provides excellent generalizability (in addition to traditional image and point cloud, etc.), as stated in Appendix Sec.D. We also expect and will conduct more applications in our future research (like multimodality, masked pre-training, interpretability, etc.).
>
> ---
> BTW, we also uploaded the codes/checkpoints/logs for CoC with global context experiment to the anonymous code link.
>
> **Thank you for your constructive suggestions and comments. We value your feedback a lot and look forward to discussing more with you during the discussion phase and in the future.**
>
> Best,
>
> Authors

---

> > ### Author Response · Authors · 2022-12-12
> > **Thank you and expect more disucssions**
> >
> > Dear Reviewer cH7t,
> >
> > Thank you for your support and helpful comments. We've tried our best to address your concerns, and we hope our responses make sense to you. Importantly, we much value your comments and would be happy to discuss more. **Although the author-engaged discussion phase will be over by today, if you have any additional questions or open discussions, please don't be hesitant to leave more comments. We are always available at all time,  to actively address any concerns or be prepared for more discussions**.
> >
> > **Your opinions are rather important for us to improve the work!**
> >
> > Thank you!
> >
> > Sincerely,
> >
> > Authors

---

### Author Response · Authors · 2022-11-22
**Updates, codes upload, additional experiments, and new promising findings**

Dear  all Reviewers:

Thanks a lot for your constructive suggestions, which help us a lot to improve our work. Here are some high-level summaries.

----

**1. Updates:** We appreciate the feedback from each reviewer. In addition to updating the figures for improved readability, we also highlighted cutting-edge clustering-based techniques in the related work section. And meanwhile, we revised the citation format and removed unnecessary font styles.

----

**2. Codes upload and additional experiments:** Thank you for the advice from Review Mp7d and 1Gqf. Experiments on removing region partition operations demonstrate that our Context Cluster is responsible for the performance, and experiments on iterative center updates show that fixed centers can provide satisfying results. We have submitted all of the codes and training logs to the anonymous link in the abstract: https://anonymous.4open.science/r/ContextCluster. We are also happy to provide other codes, such as those for visualization, to help in other people's research.

-----

**3. New promising findings:** Thanks to the suggestions from Reviewer cH7t and Mp7d, visualization results on CoC with global interactions show promising new findings: `Context Cluster can achieve "superpixel"-like clustering results at all stages, verifying our motivation and providing gratifying interpretability`.  For your convenience, we randomly selected several examples and showed the clustering results here: https://anonymous.4open.science/r/ContextCluster/uploads/rebuttal_examples.png.

------

**We sincerely appreciate all reviewers for your comments and for helping us to improve our work. During the rebuttal, our Context Cluster has shown more advantages and benefits, and we believe there are many promising potentials and interesting directions that are worthwhile exploring. Feel free to let us know if you have any further questions or concerns. Thank you very much.**

---

### Author Response · Authors · 2022-12-12
**General comment (December 12, 2022)**

We'd like to again thank all the reviewers for their constructive suggestions and for engaging in helpful discussions. Your comments on our work are much appreciated and will help the work be stronger. Below, we summarize the revisions we have made in response to the review:

-----

1. **Some major edits made in the current revision**:
    - We have revised the related work to pay more attention to the cluster-related papers, as suggested by Reviewer cH7t.
    - Thanks to Reviewer cH7t and Reviewer 1Gqf, we have updated Fig. 4 and Fig. 6 for better illustration.
    - We also updated some improper font formats, as suggested by Reviewer cH7t and Reviewer QuSd.
    - More experiments are conducted, including introducing global modeling capabilities, removing region partition operations, and iteratively updating centers. Thanks to all the reviewers (cH7t,Mp7d, Qusd, and 1Gqf).
    - Besides we also added more figures (Fig. 8, Fig. 9, and more figures in the anonymous link) to demonstrate the clustering results of our CoC with global interactions, which exhibit great superpixel-like clustering results and provide gratifying interpretability.
    - We fixed the incorrectly used citation format and unclear appendix reference, as pointed out by Reviewer Qusd.
    - We added more comparisons like Swin-T during rebuttal, as suggested by Reviewer Mp7d.

----

2. **Highlighted results and exciting findings during rebuttal**:
	- Global modeling capabilities can consistently improve the empirical performance of our method. We validate this with two new experiments, removing the region partition operation and introducing global context via global average pooling.
	- Without the limitation of local receptive field, our method exhibits promising clustering results in a superpixel-like style. This phenomenon validates our motivation and provides gratifying interpretability. See: https://anonymous.4open.science/r/ContextCluster/uploads/rebuttal_examples.png
	- Besides context-aware clustering results, our method also can learn a fixed and context-irrelevant clustering pattern, like image borders. See: https://anonymous.4open.science/r/ContextCluster/uploads/rebuttal_border_cluster_stage3.png

----

**We thank you for your comments and responses. We did our best to address the concerns raised by reviewers, and we appreciate these improvements could be considered. We will also add these results and findings to our manuscript.**

---

> ### Author Response · Authors · 2022-12-13
> **Thank you to all Reviewers, please feel free to post any further concerns or discussions (December 13, 2022)**
>
> **Dear Reviewer cH7t, Mp7d, QuSd, 1Gqf,**
>
> We thank you for all your comments and responses.
>
> We did our best to address the concerns raised by reviewers, and we appreciate these improvements could be considered.
>
> **If you have any further questions or concerns, please feel free to post, and we will continually work on this project and actively address your concerns (if any further, even after the discussion phase 2). We look forward to more discussions and your suggestions, which are important for improving the work! Thanks for your time and contributions.**
>
> Sincerely,
>
> Authors.

---

### Decision · Program_Chairs · 2023-01-20

**Decision:**

Accept: notable-top-5%

**Justification For Why Not Higher Score:**

N/A

**Justification For Why Not Lower Score:**

The paper proposes a very novel and interesting approach to deep, hierarchical image representation. It is not a variant of existing themes, and the manuscript is well-written with excellent motivations for its main contributions throughout. The experimental evaluation considers a broad variety of recognition problems (in images and pointclouds), providing convincing empirical evidence of the usefulness of CoC as a generically applicable backbone architecture.

**Metareview: Summary, Strengths And Weaknesses:**

# Summary of Contribution

This paper proposes a new visual representation learning pipeline called Context Clusters (CoC). The main idea is to consider the image as a set of points (similar to a pointcloud) and to use a clustering algorithm to form local context clusters instead of treating the image as a regular grid of patches. Point features in a cluster are aggregated using an adaptive average pooling and then "dispatched" back to the constituent points via a fully-connected layer. The architecture further considers multi-head dispatching like in self-attention networks. Experimental results are given on a variety of recognition tasks, including ImageNet-1K object recognition, 3D pointcloud classification on ScnObjectNN, object detection and instance segmentation on MS-COCO, and semantic segmentation on ADE20K.

# Strengths

+ **Novelty**: The proposed approach is novel, naturally interpretable, and shows great potential as a hierarchical model for image representation based on deformable convolutions instead of fixed-patch convolutions.

+ **Experiments**: The breadth and depth of the experimental evaluation provides very strong evidence of the efficacy of the Context Cluster approach. The considered CoC models outperform similarly-sized convolutional and Transformer-based architectures on multiple tasks and have comparable computational costs. The simple ablation study in Table 2 supports the claim that each contribution of the CoC architecture (positional coding, context clustering, and multi-head dispatch) all contribute to the performance.

+ **Interpretability and Reproducibility**: Context Clusters are as interpretable as convolutions, and the experimental results seem reproducible based on the technical details given in the paper (code is also provided). The supplementary appendices provide more detailed and complementary results and analyses.

+ **Clarity**: The paper is generally well-written and the main contributions are motivated from first principles throughout (see Weaknesses below for some qualifications).

# Weaknesses

+ **Clarity**: The paper needs some editorial revision for language and some elements of the technical development. Some reviewers point to difficult passages (some of these were addressed in rebuttal), and to difficulty in interpreting some figures -- especially those intended to support the interpretability of the model. Finally, reviewers single out several typographical inconsistencies that interfere with readability.

+ **Comparison with Other Architectures**: Reviewers pointed to the Swin Transformer as a self-attention model with similar motivations (the authors provided a comparison in rebuttal). A more measured comparison with models having similar goals is in order. The novelty of the proposed approach is not at all in question at this point, so a careful consideration of alternative approaches seems appropriate.


# Summary

The reviewers are overwhelmingly positive in their evaluation of the paper. Their concerns were addressed through a lively back-and-forth with the authors during the discussion phase, which has significantly clarified the novelty and effectiveness of the CoC architecture. What the authors propose is a novel approach to deep, hierarchical image representation that adds something interesting to the discussion on deep representation learning for visual recognition.

That said, there is definitely room for improvement. At many points in the paper the authors interject somewhat vacuous assertions about the novelty of the approach, for example restating many times that it is distinct from self-attention, which becomes a bit gratuitous. More effort could be spent meaningfully contrasting and comparing alternative architectures.


**Note From Pc:**

if the above contains the word "oral" or "spotlight" please see: "oral" presentation means -> notable-top-5% and "spotlight" means -> notable-top-25%. As stated in our emails, we are disassociating presentation type from AC recommendations